# Mycotoxin Monitoring, Regulation and Analysis in India: A Success Story

**DOI:** 10.3390/foods12040705

**Published:** 2023-02-06

**Authors:** Sujata Chatterjee, Archana Dhole, Anoop A. Krishnan, Kaushik Banerjee

**Affiliations:** 1National Reference Laboratory, ICAR-National Research Centre for Grapes, Post Office, Manjari Farm, Pune 412307, India; 2Export Inspection Agency, Kochi 68206, India

**Keywords:** aflatoxins, APEDA, FSSAI-ML, mycotoxins, method validation

## Abstract

Mycotoxins are deleterious fungal secondary metabolites that contaminate food and feed, thereby creating concerns regarding food safety. Common fungal genera can easily proliferate in Indian tropical and sub-tropical conditions, and scientific attention is warranted to curb their growth. To address this, two nodal governmental agencies, namely the Agricultural and Processed Food Products Export Development Authority (APEDA) and the Food Safety and Standards Authority of India (FSSAI), have developed and implemented analytical methods and quality control procedures to monitor mycotoxin levels in a range of food matrices and assess risks to human health over the last two decades. However, comprehensive information on such advancements in mycotoxin testing and issues in implementing these regulations has been inadequately covered in the recent literature. The aim of this review is thus to uphold a systematic picture of the role played by the FSSAI and APEDA for mycotoxin control at the domestic level and for the promotion of international trade, along with certain challenges in dealing with mycotoxin monitoring. Additionally, it unfolds various regulatory concerns regarding mycotoxin mitigation in India. Overall, it provides valuable insights for the Indian farming community, food supply chain stakeholders and researchers about India’s success story in arresting mycotoxins throughout the food supply chain.

## 1. Introduction

Mycotoxins are secondary metabolites of fungi belonging to the phylum Ascomycota [1]. These toxins are released in a variety of foodstuffs including cereals, dried fruits, spices and milk, and their exposure might cause serious hazards to human and animal health. So far, more than 500 mycotoxins have been reported as potentially toxigenic. These include aflatoxins (AFs) and ochratoxins (OTs) produced by *Aspergillus* sp., trichothecenes (TCTs) produced by *Trichotecium* sp., fumonisins (FUMs) and zearalenone (ZEN) produced by *Fusarium* sp., citrinin (CT) and patulin (PAT) produced by *Penicillium* sp. and ergot alkaloids (EAs) produced by *Claviceps* sp. [2,3]. According to the Food and Agriculture Organisation (FAO) [4], one-fourth of the world’s food grains are estimated to be infected with AFs alone, affecting 4.5 billion lives in developing nations [5]. Out of the four variants of AFs, AFB1 has been catalogued as a group I carcinogen by the International Agency for Research on Cancer (IARC) [6]. This toxin has globally posed a serious menace to public health [7], which necessitates its year-round screening and control. 

Every year, Indian agriculture faces a variety of stresses related to pest infestations, frequent fungicide applications, adverse weather, untimely harvests, hailstorms and water logging, all of which make it difficult to implement or maintain good agricultural practices (GAPs). The Indian agro-climatic landscape is also very conducive to the development of fungal infections compared to temperate regions of North America, Canada and Europe, which report relatively fewer cases of mycotoxin-induced diseases, primarily affecting humans [8,9]. After conducting numerous tests in India, it has been noted that AFB1 is the most extensively encountered mycotoxin in food, followed by AFB2, with the occurrence of AFG1 and AFG2 being almost negligible [8]. As predicted, global climate change will have a great impact on the occurrence of mycotoxins in foods and feed, posing ongoing health risks to the vulnerable population (e.g., infants and immunocompromised adults). Thus, the monitoring and control of mycotoxins, particularly AFs, in foods is extremely warranted.

Pre- and post-harvest management practices (e.g., cropping pattern, weed control and handling of grains) have potential implications, especially on the health of stored crops [10]. Mycotoxin-producing fungi associated with grains can be categorised into field fungi and storage fungi [11]. Storage fungi, chiefly Penicillium, Aspergillus and Fusarium, are responsible for the loss of 25–40% of cereal grains [12]. One key factor behind this infestation is the lack of advanced post-harvest storage facilities such as metal silos and aerated bins. Small-holder farmers in developing nations resort to jute bags, bamboo structures and baked earthen containers for storing harvested grains, that are unregulated and ill-equipped for restricting moisture and oxygen migration, providing ideal conditions for microbial attack. In North Indian farms, for instance, grain sacks weighing 50–60 kg that were stacked and cushioned by rice straw were affected by these toxins [10]. Additionally, there are problems related to inadequate drying (moisture content exceeding 12–15%) and faulty sealing. The production of various traditional Indian delicacies by small business operators (e.g., papadam made up of rice, sagoo or plant-based beans, taken both as an appetiser and a side dish) with inadequate hygienic control further accelerates fungal growth. Identifying these shortcomings and addressing them are the first steps towards implementing remedial strategies to control mycotoxins.

It is well documented that the European Commission (EC) and its Member States maintain a high level of safety and ensure quick responses to threats pertaining to food and feed. One key tool used to rapidly react to such safety crises is the Rapid Alert System for Food and Feed (RASFF). For example, in RASFF’s annual report in 2020, AFs were one of the top ten hazards detected and notified in fruits and vegetables, nuts and their products, dried figs and seeds from Turkey, the USA, Argentina and Spain [13]. Between 2017 and 2020, the number of notifications in RASFF and the Administrative Assistance and Cooperation System (AAC) revealed a rapid rise in product non-compliance. During this time period, fruits and vegetables accounted for the majority of non-compliance notifications (15%). Strikingly, there were 367 notifications for AFs, particularly for dried figs (58 notifications), groundnuts (29 notifications) and feed (22 notifications). OTA was mostly detected in fruits, vegetables and nuts, mainly dried figs [13]. By 2020; however, there had been a steady decline of mycotoxin notifications by 23%, particularly in peanuts and their processed products. 

Evidently, AF poisoning is one of the primary causes of hepatocellular carcinoma, kidney disorders, neonatal jaundice and endocrine disruption in resource-limited countries [14]. One of the earliest instances of AF poisoning in India dates back to the 1970s, when people from almost 200 villages in the states of Gujarat and Rajasthan reportedly suffered from hepatitis after consuming mouldy maize [15]. Fast forward again to recent times, a group of Indian researchers [16] reported the presence of Aflatoxin M1 (AFM1, a metabolite of AFB1 found only in milk) in 41% of human milk samples and 93% of animal milk samples in a risk assessment study from Haryana. They also noted that women who were predominantly on a rice and flour diet exceeded AF’s Provisional Maximum Tolerable Daily Intake (PMTDI) limit [16]. Obviously, the study revealed that infants are in a high-risk situation, as this age group is heavily dependent on dairy products for nourishment. Such a finding indicates that the rate of child mortality, the loss of valuable lives or other health complications might considerably increase if no timely measure is taken.

On the brighter side of things, the food safety landscape in India has witnessed a wholesome change over the past two decades. Two nodal governmental agencies, namely the Agricultural and Processed Food Products Export Development Authority (APEDA) and the Food Safety and Standards Authority of India (FSSAI), have taken actions and updated regulations to monitor safe mycotoxin levels in widely consumed food products. The maximum limits (MLs) are now largely harmonised with the standards prescribed by the Codex Alimentarius Commission (Codex) and other regulatory bodies of importing markets. Additional export control and surveillance agencies, including the Spices Board (SB) and the Export Inspection Council (EIC), both established by the Ministry of Commerce and Industry, are monitoring economically significant commodities such as ethnic spices and nuts (and their processed products) that are highly popular overseas. 

In the past few decades, food laws have been subjected to numerous revisions as per novel testing requirements. For the customs clearance of food consignments, an Indian food business operator must comply with various governmental regulations to procure a health certificate. To keep up with the ever-evolving trade requirements, Indian regulatory agencies, in cooperation with analytical scientists at the National Reference Laboratory of the Indian Council of Agricultural Research’s (ICAR) National Research Centre for Grapes (NRL-NRCG) in Pune, have introduced advanced testing methodologies involving high performance liquid chromatography (HPLC) with fluorescence and mass spectrometry (MS)-based detection mechanisms. Since 2004, NRL-NRCG, an APEDA-nominated laboratory in Pune, has developed several high-throughput mycotoxin analysis methods. A system like this not only controls food quality but also limits trade failures. Even though a great number of reviews have focused on AFs in the past five decades, comprehensive information on such advancements in mycotoxin testing and issues in implementing these regulations in the Indian context is inadequately covered in recent literature. Although these advanced testing methods have been widely adopted by food testing laboratories across the nation, they are scarcely covered in a review. 

Taking cues from the above-stated gaps, the present review upholds a systematic picture of the role played by the nodal Indian governmental agencies (e.g., the FSSAI and APEDA), through coordinated efforts of capacity building, disseminating knowledge, training laboratory professionals, and creating traceability platforms in mycotoxin control, both at domestic and international levels. Additionally, it describes how these agencies have established a food safety network comprising primary, referral and reference laboratories that are actively involved in sampling and mycotoxin testing using validated methodologies. The functions of web-based traceability platforms (e.g., Peanut.net of the APEDA) are also highlighted, with an emphasis on mycotoxin regulation. Furthermore, it elucidates recent regulatory changes and alternative approaches to managing mycotoxin in India. Critical remarks on selected approaches, their validation parameters and applications are provided to help readers appraise the significance of these breakthroughs. This review, the first of its kind, provides valuable insights into India’s mycotoxin monitoring and testing systems, capacity building, challenges in implementing legislation in food export and unexplored food commodities that require future attention and will be of interest to a wide range of readers.

## 2. Methodology

We referred to MDPI Publications’ “Instructions for Authors” guidelines to prepare this review article. Recent research and gaps in India’s mycotoxin surveillance led us to determine the review’s aim and scope. Initially, relevant papers were selected by screening the “Title” and “Abstract” of published materials. First, all downloaded citations were exported to Microsoft Word and annotated as footnotes; these were then added in the reference section following the referencing style of MDPI Foods.

We searched for relevant data to:Understand how monitoring and controlling mycotoxins in foods influences public health.Determine why Indian agricultural practices are prone to fungal infection and the generation of mycotoxins in grains, nuts, spices and milk.Illustrate India’s food safety framework for mycotoxin surveillance along with the ML of major regulating countries.Enumerate strategies acquired by the export monitoring agencies of India (APEDA, EIC and SB) for reducing consignment rejections and enhancing the trade of spices and nuts.Elucidate the fundamental role played by governmental and private laboratories in mycotoxin testing. Describe the analytical methods developed by NRL-NRCG.Identify challenges associated with regulation and the implementation of mitigation programmes.Provide an overview of the impacts of these regulations on Indian trade and commerce. Identify gaps in global safety standards and legislation for mycotoxins.

### 2.1. Study Design

Google Scholar, Scopus, Wiley Online Library, ResearchGate, PubMed and Academia.edu were used to conduct electronic searches. Furthermore, the Google and Yahoo search engines were used to find key international and federal entities linked with mycotoxins and related food safety standards. Additionally, we contacted regulatory specialists to gather authentic information on AF regulations and the latest MLs.

Specific web-based resources (not exhaustive) were also referred to:FSSAI;EU;APEDA;SB;EIC;The National Dairy Development Board (NDDB), Anand;National Medicinal Plants Board;The World Health Organisation, Food and Agriculture Organisation of the United Nations, The Joint FAO/The World Health Organisation (WHO), Expert Committee on Food Additives (JECFA) and Codex;United States Food and Drug Administration (USFDA);RASFF; andTrilogy Innovations Private Limited, Hyderabad.

### 2.2. Inclusion Criteria

Only publications written in the English language were considered, and an effort was made to include recent literature, although certain original papers were referred to while citing old findings. Only a few studies available online in regional languages (Marathi and Hindi) were considered. Analytical methods were included while keeping in mind the regulatory requirements of India and its trading countries.

### 2.3. Exclusion Criteria

Any publications with specific mycotoxins other than AFs, OTs, PAT, FUMs, T-2 toxin and ZEN were excluded. Articles that appeared in low-impact journals and did not address the Indian regulatory framework were disregarded. Information sourced from unofficial webpages was excluded.

## 3. India’s Food Safety Framework for Domestic Surveillance of Mycotoxin

The development of fungal infection and mycotoxin production are climate-sensitive processes, and because India has a tropical climate, mycotoxins in food commodities are likely to flourish. Given that there are notable seasonal and spatial variations in mycotoxin contamination levels in the country, their detection rates vary. Handling and storage conditions are important risk factors associated with post-harvest mycotoxin accumulation. The extent of AF production in cereals depends on temperature, moisture, soil type and storage conditions. Among cereals, rice is mostly contaminated with AFs in the natural environment. While nuts are highly susceptible to AF contamination both in the field and during storage, spices are significantly affected by storage and processing conditions [17]. For this, the Indian food industry has implemented several internal monitoring mechanisms for domestic control. 

Mycotoxin surveillance in food and feed is conducted on a regular basis by the country’s nodal regulatory agencies as part of pre-export testing. These control methods are mostly preventive in nature, including the implementation of GAPs at the field level and adequate crop drying after harvest. The International Crops Research Institute for the Semi-Arid Tropics (ICRISAT) in Hyderabad, for instance, is conducting extensive research into methods to prevent crop contamination before harvest. Many efforts simply comprise removing mycotoxin-contaminated goods from the food supply chain via governmental screening and regulatory programmes. To reduce mycotoxin contamination, governmental agencies have implemented advanced agricultural technologies such as GAPs, good manufacturing practice (GMP) and good storage practice (GSP). The crop-specific institutes under ICAR and the state agricultural universities have published numerous GAP-based pre-harvest guidelines (e.g., fertiliser and pesticide management, weed management, water and nutrient management, among others). Numerous post-harvest techniques (e.g., drying, maintaining proper moisture levels, sorting damaged or infested grains and destroying contaminated crops) are adopted, too. Additionally, biological control has been proposed as a possible method for limiting the growth of mycotoxin-producing fungal communities [18]. However, strict adherence to these practices is still required.

In the following paragraphs, we depict how the Indian government is instituting a hierarchy with respect to food testing laboratories in India to enhance accountability and transparency in its food testing systems.

### 3.1. Role of FSSAI in Capacity Building

The FSSAI is the chief regulatory body responsible for building an effective food safety network to provide scientific information to consumers regarding safe and healthy food. On 20 October 2020, this agency published a revised manual on mycotoxin analysis [19], where MLs of all concerned toxins in food matrices were compiled [20]. These include Afs in peanuts [21,22], deoxynivalenol (DON) in wheat [23], ZEN [24] and FUM [25] in maize, and PAT in apple juice [26]. It made the manual available to the public to facilitate foreign trade.

In accordance with Section 43 of the Food Safety and Standards (FSS) Act of 2006, the agency has granted approval to several food testing facilities that are accredited to the ISO/IEC17025:2017 standard. Mycotoxin levels in local and imported food items are continuously monitored in three national food laboratories (NFL), one of which is located in Ghaziabad and the others in Kolkata and Mumbai [27,28]. Under the FSS Act of 2006, these apex food laboratories are responsible for food quality testing. The NFLs in Ghaziabad and Mumbai are operated through public–private partnerships. There are 227 primary food laboratories accredited by the National Accreditation Board for Testing and Calibration Laboratories (NABL) [29], which are meant to cover all zones (Northern, Southern, Eastern and Western) of the country. 

Furthermore, FSSAI has recognised 19 referral laboratories (RLs) for analysing appeal samples that are rejected on account of above-ML residue issues [30]. In addition, 11 national reference laboratories [31] have been recognised for establishing nationwide standards for routine analysis, developing new and reliable methods, validating such testing methods and organising proficiency testing (PT). These laboratories are part of the most prestigious research and development organisations in the nation and are located in Mysore (1), Mohali (1), Kochi (3), Pune (1), Anand (1), Lucknow (1), Hyderabad (1), Kolkata (1) and Gurugram (1). Additionally, two ancillary laboratories in Chennai and Kolkata have been identified to assist the NFLs in organising PTs. Whereas, National Reference Laboratories, located at NRL-NRCG and Trilogy Analytical Laboratory Pvt. Ltd., Hyderabad, have been assigned for mycotoxin analysis and providing reference materials, respectively. Both of these laboratories offer support in analysing imported samples, particularly when a primary food laboratory rejects them for having mycotoxin contamination levels exceeding the national ML. A schematic representation of the FSSAI’s role in capacity building is provided in Figure 1.

Concurrently, the FSSAI’s Scientific Panel on “Methods of Sampling and Analysis” makes insightful recommendations on existing testing methods and incorporates new test methods for analysing new parameters (commodity-ML combinations). The agency also provides courses on the microbiological and chemical aspects of mycotoxin testing to laboratory personnel from FSSAI-notified and central food laboratories, among others [32]. In 2019, two such “Training of Trainers” programmes on mycotoxin analysis took place at NRL-NRCG and the Export Inspection Agency (EIA), Kochi [32]. Additionally, the FSSAI provides portable lateral flow assay readers to all regulatory laboratories in various states and union territories to facilitate mycotoxin monitoring. This action was taken while keeping in mind that biological toxins can cause permanent health damage.

Under Section 16 (2) (f) of the FSS Act, 2006, the agency has published specific manuals on analytical methods for different food commodities with a “one parameter–one method” approach. The first mycotoxin analysis manual, released in 2016, was written with the primary goal of providing thorough and up-to-date methodologies for regulatory compliance. After several revisions, the latest version was published in 2021, in which several validated methods (mostly contributed by NRL-NRCG) of AFs, OTA and PAT were incorporated. This analytical manual summarises regulatory limits (Table 1), safety requirements and 22 well-defined analytical methods comprising sample preparation as well as instrumental analysis summarised in Table 2. By making these records accessible to the public, the FSSAI has significantly improved current mycotoxin testing practices.

### 3.2. Risk Assessment-Based Surveillance 

Risk assessment is a conceptual framework that aims to measure the potential for adverse health impacts from foodborne illness. As previously stated, AFs are known to influence a wide range of cellular processes and exert toxicological effects. To avoid such risks, a health risk assessment organisation must be constituted with the task of anticipating the health implications of certain contaminants (e.g., mycotoxins) over time. In this regard, the FSSAI has established a risk assessment cell (RAC) under Section 10, 16 (1) (i) (c) and Section 18 (1) (2) (b) (c). The RAC performs risk assessments to facilitate risk management and communication among the stakeholders involved. Additionally, it identifies hazards that contribute to foodborne health problems besides collecting data on contaminants from research and development projects (pull and push types) and scientific panel findings. Food safety officers across the country collect samples, which are then tested in FSSAl-regulated laboratories. The values are quantified and assessed for risk by comparing them with MLs. The results are then presented to risk managers and stakeholders [33,34]. Based on such risk assessments, the FSSAI has recently reduced the limits of total AFs from 30 μg/kg to 15 μg/kg and set limits for AFB1 at 10 μg/kg in cereals and nuts [19,35,36]. By reducing mycotoxin MLs, India has minimised long-term health risks to its consumers. However, more emphasis should be placed on the risk assessment of mycotoxin consumption in meals (say, AFB1 in peanuts, AFM1 in milk) to examine health risks to the Indian population. Therefore, a multidisciplinary effort should be developed to assess the human health risk of multiple mycotoxins found in food, with risk managers prioritising risk and developing disease prevention actions based on the information gathered from the risk assessment process. Risk assessment of numerous mycotoxins is challenging, requiring expertise from multiple scientific domains as well as significant effort from the scientific community, risk assessors, and managers [37]. 

Having addressed the FSSAI’s pivotal role in domestic mycotoxin monitoring, we will now go on to describe how monitoring authorities respond to rapid alerts, followed by the roles of APEDA and the NRLs in testing, validation and export facilitation.

## 4. Mycotoxin Surveillance for Export Facilitation

As previously stated, RASFF is a critical tool for quickly responding to food safety crises, because it notifies the public when an imported consignment fails the European food legislation. The European Union (EU) has defined rules [Commission Implementing Regulation (EC) No.2019/1793] on the frequency of identity and physical checks for food consignments entering from India. For instance, 10% of consignments (e.g., rice, whether husked, semi milled or wholly milled) are randomly checked at the port of entry to the EU for AFs and OTA. Whereas, 50% of peanuts and their processed product shipments are subject to rigorous AF testing. In 2020, two AF-related RASFF notifications were reported for rice, 10 notifications for herbs and spices and 19 notifications for peanuts and their processed products. In response to an increasing number of RASFF notifications of AF contamination in peanuts imported from India, the EC/Directorate-General for Health and Food Safety has recently conducted certain inspections.

It is an established fact that trading agricultural commodities is a means of revenue generation and economic progress. During 2020–2021, India exported 680,000 tonnes of peanuts, and some of its top importers were Indonesia, Vietnam, China, Malaysia, Nepal, Russia, the Philippines and Thailand [38]. Like peanuts, other nuts, spices, medicinal herbs and milk are rich substrates for fungal attack, and hence it is imperative that rigorous surveillance be in place to minimise product recalls upon reaching the importing countries. To accomplish this, the government has introduced the following schemes, platforms and monitoring programmes to oversee the export of these food matrices, as described below.

### 4.1. APEDA—Peanut.Net

As an export development organisation established by the Ministry of Commerce and Industry (under the APEDA Act of the Government of India) in 1985, APEDA has launched several traceability platforms. These include HortiNet to monitor the export of fresh fruits (e.g., grapes, pomegranates, mangoes and oranges) and vegetables (e.g., okra and green chillies) [39], TraceNet to supervise organic food export [40] and Peanut.Net to control Afs in peanuts and peanut products [41]. 

Peanut.Net is one of the most recent initiatives launched by APEDA, and its goals include increasing supply chain transparency for peanuts, ensuring compliance with AF MLs and adhering to the quality parameters established by importing countries. As of April 2022, 26 laboratories are designated for the EU, Singapore, Malaysia and the Russian Federation, while 14 laboratories are solely meant for supporting export to Indonesia [42]. For this purpose, an elaborate manual on export procedures for peanuts and peanut products has been laid down [43]. Stakeholders can apply for a Certificate of Export (COE) through Peanut.Net by registering their peanut processing units or warehouses, shelling units and grading units. Other services that can be availed of through this platform include AF analysis by NABL-accredited laboratories, monitoring by NRLs, consignment creation, reimbursement of laboratory testing fees for export and verification of the registered code for export. This online platform allows access only to those ISO/IEC17025:2017 accredited laboratories that performs well in the AF PT rounds under the supervision of NRL-NRCG. 

Sampling and AF analysis are two of the most important aspects of consignment testing that are also performed on this platform. Importantly, the exporter and his export facility need to be registered with APEDA. Sampling is performed in compliance with Commission Regulation (EC No. 401/2006), in which an NRL-trained and authorised sampling representative visits the facility and collects representative samples. Thereafter, the samples are sent to an approved laboratory, where they are evaluated for AFB1, AFB2, AFG1 and AFG2 using HPLC-fluorescence or liquid chromatography tandem mass spectrometry (LC-MS/MS). When a sample meets the ML of the importing country, the exporter receives an export code (from the office of the Director General of Foreign Trade) and a health certificate (from EIC). The lot or sub-lot can only be exported if the test sample meets the MLs after adjusting the concentration for recovery (%) and measurement uncertainty. On the other hand, a product is rejected when a test sample exceeds the ML beyond a tolerable point, which is generally within ±50% measurement uncertainty. 

In this way, APEDA is ensuring that the AF contamination level of Indian-origin products remains within the MLs of various trading countries. Most importantly, it makes use of this platform to provide e-monitoring for tracing and tracking shipments, reducing the time-consuming paperwork associated with tracing.

### 4.2. Role of NRL 

The APEDA has nominated NRL-NRCG for regulating mycotoxin testing in export consignments [44], which in turn recommends that the commercial food testing laboratories be approved based on the physical assessment of their method validation records and performance in PT rounds. The candidate laboratories must validate analytical methods in accordance with EC No. 625/2017, with special reference to the performance criteria stated in Commission Regulation (EC) No. 401/2006 [45] and SANTE/12089/2016 [46]. These validated procedures assist laboratories in maintaining the accuracy and precision of their mycotoxin analyses, thereby contributing to the integrity of the data.

As an advanced procedure of inter-laboratory comparison (ILC), a PT round allows external quality control and assurance. A laboratory can demonstrate its analytical competence by participating in a test conducted by an ISO 17043-accreditated PT provider (PTP), e.g., NRL-NRCG, who is an acclaimed ISO/IEC 17043:2010-accreditated PTP since many years. Every year, it organises at least two PT rounds on AFs in peanuts, the details of which are posted on the NRCG’s website [47] well in advance. This enables the participating laboratories to identify systematic gaps in their measurements, thereby improving their analytical capacity. Upon completion of a PT, the successful laboratories (with z scores within ±2) are informed and provided with Peanut.Net login access. On the other hand, the non-compliant laboratories undertake root cause analysis and report the same to the PTP in accordance with ISO 17025 standards. The accuracy of their results is then assessed by comparing them with the assigned value of the prior PT (reference value). If the results that they produce are reliable, a certificate of compliance is issued to them. Others are disqualified, as their results fall outside the reference value range. They may, however, reapply for the next PT round whenever it is announced. The PTP evaluates 5% of the quality control samples randomly collected from testing laboratories as a routine quality assurance procedure. This allows for a better insight into the analytical concerns that laboratories confront and aids in the resolution of non-compliance issues in order to improve mycotoxin quality testing.

### 4.3. Mycotoxin Regulations and Spice Commerce

The majority of spices sold on the global market come from developing countries, many of which have climates conducive to fungi growth. They are typically sold in the marketplace either in their whole, ground or mixed-ground forms. In Asian cuisines, a wide array of spices (e.g., turmeric, chilli, cumin, cinnamon, clove, black and white pepper, star anise, cardamom, fennel and coriander seeds) are frequently used [48]. As per the World Trade Organisation (WTO), India produces a substantial quantity of its ethnic spices, close to 3 million tonnes per year [49]. The majority of these spices are grown by small-scale farmers with limited infrastructure, and they are often processed in an open environment where exposure to fungal growth is unavoidable. Indian spices are most commonly contaminated with AFs and OTA [50], and since they play a crucial role in the national economy, the spread of mycotoxins must be arrested to prevent human health risks and non-compliance with food safety standards. Despite the immense popularity of Indian spices (e.g., pepper, cardamom, dried ginger, turmeric, cumin, fennel and chillies) across the world, the spice trade is hardly realising its true potential. For instance, Reddy et al. (2010) investigated AFB1 contamination in fresh chillies and chilli powders from markets in Andhra Pradesh and noted that AFB1 was detected in 59% of the chilli samples, with fresh chilli having the highest concentration (969 μg/kg) [51]. They also detected AFs in nearly 9% of the chilli powder samples drawn from supermarkets. In a 2015 study by Jeswal and Kumar, Aspergillus flavus was the most prevalent in black pepper among four others, namely green cardamom, turmeric, mace and fennel [52]. All this suggests implementing preventive measures for controlling mycotoxin contamination in spices.

To promote and increase the sale of high-quality Indian spices in international markets, SB was founded in 1986 by the Union of Cardamom Board and the Spices Export Promotion Council. Currently, it is managed by a network of Quality Evaluation Laboratories (QELs), located in Kochi, Mumbai, Kandla, Kolkata, Guntur, Chennai, New Delhi and Tuticorin. It lays down mandatory sampling and testing protocols for export consignments under its Quality Evaluation System, in which MLs of AFs are listed corresponding to the importing countries [53]. All laboratories under the purview of QELs are well equipped with the necessary instrumentation for AF testing [54]. Through its annual training sessions at the Kochi Centre, the board also disseminates information on the analysis of mycotoxins and illegal dyes in spices and spice products. As a result, food analysts develop their expertise in accurate mycotoxin extraction and testing, eliminating handling errors and reducing matrix effects.

Most spices are processed before being consumed or marketed. Because most of the mould proliferation occurs during processing, FSSAI published a detailed Food Industry Guide in 2018 as part of its Food Safety Management System (FSMS) for implementing Good Handling Practices (GHP)/GMP requirements during spice processing. Similarly, the EU has mandated the ML for AFB1 to be 5 μg/kg and total AFs to be 10 μg/kg in spices including chilli, peppers, nutmeg, ginger, turmeric and a spice mixture [55]. Alongside these MLs, several parameters, including the number of dead insects, foreign objects, mammalian waste and the percentage of spoilt items, are critically inspected. The exported consignments must thus be periodically evaluated to check whether they qualify for the Defect Action Level of the USFDA and comply with the cleanliness mandates set by the American Spice Trade Association (ASTA). 

Despite making such commendable advances in capacity building, the nodal regulatory bodies constantly face issues with the implementation of regulations and the management of mycotoxins at pre- and post-harvest levels. The food business operators often get confused as to why shipments are rejected at EU markets, even though they meet all export control requirements. This is primarily the result of the uneven distribution of mycotoxins within a consignment as well as the possibility of an increase in the toxin concentration at any point along the supply chain. To keep exporters from losing business, better transparency in the regulations and effective management decisions would be necessary.

## 5. Evolution of Mycotoxin Analysis Techniques

It is worth noting that Indian scientists and food analysts have adapted to fast-changing testing methodologies. Until 2006, mycotoxin testing in India was primarily carried out through immunochemical approaches such as the enzyme-linked immunosorbent assay (ELISA). However, ELISA-based approaches are not only time-consuming [56] but also suffer from a lack of selectivity due to matrix interferences. These matrix effects can lead to both false negative and false positive results. With technological progress, the EU came up with a regulation (EC 401/2006) to check domestic and imported food products for mycotoxins. In response, several Indian laboratories adopted the quality control criteria of EC 401/2006 to meet the compliance requirements of the exports of peanuts and their processed products as well as plant-derived commodities. The regulation prescribes HPLC analysis with fluorescence and/or LC-MS/MS detection for selective and accurate testing. NABL-accredited laboratories are expected to implement these prescribed methods of NRL-NRCG or the official methods of Association of the Official Analytical Chemists (AOAC) INTERNATIONAL. Today, NRL-NRCG has been instrumental in the advent of innovative analytical methods for a wide array of food products and has improved many existing AOAC Methods to meet the specific requirements of Indian food matrices before implementing them for routine analysis after appropriate validation. Due to their simple workflow, selectivity, sensitivity, ruggedness and cost-effectiveness, these methods have been successfully adopted by numerous food testing laboratories across the country. In this manner, NABL-accredited mycotoxin testing laboratories save time and money on method development by using fit-for-purpose methods readily available to them.

When FSSAI became operational in 2011, the MLs for seven classes of mycotoxins (commonly found in various Indian foods) were gazette notified. Out of these, except AFs and OTA, the other mycotoxins were not amenable to fluorescence detection (FLD). This is how the use of LC-MS/MS came into practice, and since then, it has been a reliable tool for multi-class, multi-mycotoxin analysis. To further consolidate the food testing mechanisms, the agency signed a Memorandum of Understanding (MOU) with the AOAC INTERNATIONAL at the AOAC-India Section’s fifth annual conference in New Delhi [57]. Through this MoU, AOAC granted the country’s top food regulator unfettered access to the official methods of analysis. This made the AOAC’s Official Methods applicable to all regulatory testing and dispute resolution purposes in the country. The signing of the MOU also aided laboratories in harmonising analytical procedures, which would have otherwise required extensive time, resources and a workforce.

As analytical methods evolve, the FSSAI has been updating its manuals for testing more complex food matrices. In general, the FSSAI only includes a method in its manual after its Scientific Panel reviews and approves its single laboratory validation and inter-laboratory validation (ILV) data. For a new method to be accepted, its performance in terms of recovery, intra-laboratory precision and reproducibility in various test matrices must be submitted, ideally in accordance with the analytical quality control criteria that are framed based on EC 401/2006 and SANTE/12089/2016 guidelines. During an ILV, at least seven laboratories are expected to perform successfully. The new method should be sensitive enough so that its limits of quantification (LOQ) are lower than the corresponding MLs. The existence of a stringent laboratory framework for mycotoxin testing on a national scale is a testament to India’s remarkable progress in this area.

Following this, we turn to discuss novel methodologies developed using advanced instrumentation.

### 5.1. Testing of Mycotoxins in Peanuts and Cereals

For close to two decades, the NRL-NRCG has significantly contributed to the development of multiple methods for mycotoxin testing in export commodities (e.g., peanuts, cereals, spices and various processed products). The first published method from this organisation targeted a direct analysis of AFs (B1, B2, G1 and G2) without involving any post-column derivatisation [58]. This method highlighted the need for developing a slurry of the sample to provide satisfactory extraction efficiency and improved precision in comparison to dry homogenisation. In brief, peanuts were homogenised after adding water (1:1). The method workflow involved adding methanol-water (80:20) to a portion of the slurry, extraction, followed by immunoaffinity column (IAC) cleanup through AF-specific cartridges and finally, analysis by ultra-HPLC-FLD (UHPLC-FLD). In the estimation of AFB1 and AFG1 using a regular FLD flow cell, an additional step of derivatisation is required to compensate for their limited fluorescence properties and provide the desired method LOQs. The use of a large volume flow cell (13 μL) in the instrumentation method used by NRL-NRCG could avoid the requirement of any post-column derivatisation, which is otherwise practised in conventional methods. This standardised and validated method was successfully evaluated in peanut-processed products and selected cereals (e.g., millets, rice and corn). Subsequently, the scope of the HPLC-FLD analysis was extended to include OTA using the same method workflow, involving an IAC that was selective to both AFs as well as OTA [59]. The excitation wavelength was switched from 365 to 333 nm after the elution of all AFs to suit the requirement of OTA. This allowed high-sensitivity estimation of all AFs and OTA in a single HPLC run with selectivity established through compound-specific excitation-emission wavelength combinations. This was a watershed development in the analytical sciences, as both AFs and OTA can now be simultaneously determined, saving time and resources as compared to individual analysis.

### 5.2. Testing Apples for Patulin

Apples are nutrient-dense fruits that include a variety of phytochemicals and dietary fibres. According to FAO, apples and value-added apple products are the 17th most-produced commodity in the world [60]. From April to December 2021, apple imports into India more than doubled and were 14% greater than the previous fiscal year [61]. The most frequent postharvest pathogen in apples is *Penicillium expansum*, which manifests in secretion of its metabolite, PAT [60]. Despite PAT being included in the FSSAI food standards, earlier editions of the mycotoxin manual did not feature any confirmatory methods for analysing it. In 2021, NRL-NRCG reported a robust and sensitive method for the LC-MS/MS analysis of PAT in apples and apple juice [26]. The method involved extracting homogenised samples (10 g) with ethyl acetate (10 mL), cleaning with dispersive-solid phase extraction using primary secondary amine (25 mg/mL) and LC-MS/MS analysis within a runtime of 5 min. It was highly sensitive, with an LOQ of 0.005 mg/kg. The method reproducibility was also established through an ILV study involving 13 accredited laboratories. Because this method has been included in the newly published FSSAI mycotoxin manual, all official control laboratories are currently able to test the domestic and import shipments in a time-efficient manner.

Another commercially notable agricultural commodity in India is mango. In a recent study from Pakistan, Hussain et al. noted a high level of PAT in a mango sample (6415 μg/kg) [62]. Similarly, there have been multiple cases of mycotoxin detection in mangoes, originating from other countries. Even if there have been no rapid alerts or detentions of Indian mango consignments overseas, it would be important to validate the method of patulin analysis in mangoes and mango juice to prevent any future trade issues.

### 5.3. Multimycotoxin Detection in Chilli Powder 

Worldwide, chilli (*Capsicum annuum*) is used as a flavouring agent that also adds colour to foods. The natives of India prefer chilli in all of its forms including fresh, dried, powder and paste. The primary source of AF and OTA contamination in chilli is the sun-drying process, which is usually carried out by spreading the mature pods on soil. The chance of contamination increases when the drying process and storage take place in hot and humid conditions [63]. Over the last decade, the Netherlands, Poland, Cyprus, the UK and Greece rejected red chilli powder consignments from India on grounds of excess AFs and OTA [64]. Repeated rejections could lead to specific import requirements or even the banning of shipments from India. To address this, the NRL-NRCG has recently published a multi-mycotoxin analytical technique that enables the simultaneous assessment of all 4 AFs and OTA in chilli powder. Usually, the analysis of either of these toxins typically takes two different approaches, but, in this case, the multi-mycotoxin immunoaffinity cartridge (AFLAOCHRA PREP) provided excellent sensitivity, accuracy and precision in a single workflow [65]. The sample preparation workflow starts by extracting powdered samples (25 g) with methanol-water (100 mL, 80:20). An aliquot (3 mL) is cleaned on an IAC and analysed in a single chromatographic run utilising UHPLC-FLD and LC-MS/MS techniques. The method performance was assessed using intra- and ILV investigations as well as against a certified reference material. As a direct result of the elimination of two separate methods of analysis, the testing has become more expedient and more cost-effective. This method will most likely be very useful to horticulturists, processors, traders and consumers alike to keep mycotoxins out of the food supply chain.

### 5.4. Automated AF Testing in Rice and Peanuts

Food testing laboratories around the world are overburdened with fresh and processed product samples, and any delay in evaluating perishable items can compromise their shelf life and economic value. Supply chain stakeholders must closely adhere to food legislation and receive all relevant certifications to encourage international trade. In India, food testing laboratories receive an average of fifty or more samples per day, making it difficult to complete the analysis in just one to two days. 

The greatest bottleneck in mycotoxin analysis is IAC cleanup, which takes a long time and requires a committed human resource for operation and supervision. For the first time in India, at the NRL-NRCG, Dhanshetty, Thorat & Banerjee optimised, developed and validated a semi-automatic method for analysing AFs in rice, peanuts, their processed products and sorghum [66]. In this method, the IAC cleanup and HPLC analysis were performed automatically, while the extraction step was performed manually. This method utilised a special kind of IAC cartridge, which could be reused 15 times without a loss in recovery. Automated cleanup and HPLC analysis provided higher precision (RSD, ≤10%) than traditionally practised cleanup (RSD = 12–15%). The main advantage of the method is that it reduced the time and cost of analysis, sample manipulation and solvent consumption, allowing for high-throughput analysis and precise results. Another advantage is that it supports human health by reducing the exposure of laboratory personnel to hazardous chemicals. Additionally, automated mycotoxin testing protocols should be developed and validated for other commodities. As the high costs associated with automation hinder Indian laboratories from adopting it, automated methods have not yet gained widespread acceptance, although this is expected to change in the near future.

### 5.5. Testing of Medicinal Plants and Botanicals

India’s long association with traditional medicine, in particular Ayurveda, is widely acknowledged [67]. The overall market demand for raw herbal pharmaceuticals for the fiscal year 2014–2015 was 5,12,000 MT [68]. Unfortunately, herbal products are frequently contaminated with mycotoxins, which puts the elderly population at risk because they consume a lot of health-boosting supplements manufactured from such medicinal herbs [69]. Previously, Roy et al. (1988) reported 14 out of 15 Indian medicinal plants to be positive for AFB1 at levels ranging from 0.1 to 1.2 μg/kg [70]. Returning to recent times, Aiko and Mehta tested 63 Indian medicinal herbs in 2016 and found AFs and CT in 33% of them [71]. These plant-derived health supplements are an essential component of Asian tradition because they contain a high concentration of phenolic compounds, which are known to possess anti-inflammatory, anti-cancer and antioxidant properties. In comparison to grains and nuts, these herbs are less susceptible to mycotoxin contamination [72]. There have been recent incidences of medicinal herbs being contaminated with emerging mycotoxins, namely beauvericin (BEA) and enniatins (ENs). As neither of these toxins has specified MLs, which is a point of regulatory concern, their testing for mycotoxin contamination assumes high significance. 

In India, the consumption of medicinal plant species, namely tulsi (*Ocimum sanctum*), ashwagandha (*Withania somnifera*), safed musli (*Chlorophytum borivilianum*), satavari (*Asparagus racemosus*) and giloy (*Tinospora cordifolia*), for healing purposes has been a customary practice. Many of them, especially giloy, possess anti-COVID-19 properties and its processed products have come under increasing attention in the past two years [73]. NRL-NRCG made an important breakthrough in developing a multi-mycotoxin method for these five prominent Indian medicinal plants. By detecting nine regulated mycotoxins [74], the simple, robust and precise method using HPLC-FLD and LC MS/MS techniques can be widely adopted for regulatory testing purposes in herbal supplements.

### 5.6. AFM1 Detection in Milk

Milk is a valuable commodity that runs the risk of being contaminated with AFM1, a major toxin that poses a threat to public health. As an instance, a study was conducted in all districts of Punjab to examine the prevalence of AFM1 in bovine milk. Among the 402 samples evaluated (266 cow milk samples and 136 buffalo milk samples), 56.2 and 13.4% exceeded the EU (0.05 μg/kg) and FSSAI (0.5 μg/kg) MLs for AFM1 in milk respectively [75]. As a hydroxylated metabolite of AFB1, this toxin is relatively resilient to detoxification attempts such as sterilisation, pasteurisation and boiling [76]. AFM1, like AFB1, is classified as a Group I human carcinogen by IARC [77]. Given the threat it poses to humans, the EU also requires that AFM1 concentrations shall not exceed 0.05 μg/kg in milk and 0.025 μg/kg in infant milk-based products [77], whereas the USA sets this limit at 0.5 μg/kg and 0.025 μg/kg respectively [77]. Furthermore, Codex has mandated that AFM1 levels in bulk milk shall not exceed 0.5 μg/kg [75]. The above variations in ML across regulatory agencies call for harmonisation to improve trade regulation and risk assessment [78].

In India, AFM1 is mostly analysed by using FSSAI 07.013:2020 and FSSAI 07.014:2020, which are both derived from AOAC Official Methods 986.16 and 2000.08 respectively [19]. In collaboration with certain industry partners, NRL-NRCG developed a high throughput method based on atmospheric pressure-matrix assisted laser desorption/ionisation (AP-MALDI) to identify this toxin in milk [79]. Such cost-effective and time-saving analytical techniques have the potential to transform existing laboratory capacities by generating accurate results in a short span. However, this technique is expected to gain momentum in promoting the development of mycotoxin testing. 

AFM1 in milk gets generated in animal bodies through the metabolic conversion of AFB1. Since feed is the main source of AFB1, it is equally important to monitor the AFB1 level in feed materials [80]. A recent paper from the NRL-NRCG has demonstrated how AFs can be estimated in a wide range of animal feeds by UHPLC-FLD without a step of derivatisation [81]. The method is sensitive enough to comply with the maximum permissible limits for AFB1 in feed (20 μg/kg) and M1 in milk (0.5 μg/kg). However, more such studies of AFB1′s occurrences in feed are desired from various corners of the nation from the risk assessment and risk management points of view. The NDDB is extensively working in this direction, monitoring the prevalence and control of AFs in dairy feeds and milk [82].

After discussing innovative techniques for detecting AF levels, we now focus on how alternative initiatives are being implemented in the country to reduce the concentration of these toxins in foods while preserving their nutritional and palatable values. 

## 6. Alternative Control and Government Initiatives

The aforementioned discussion conveys the growing AF threats that are focused on commercial food products; however, there must be alternative strategies to make the country’s citizens aware of mycotoxin contamination, especially when they access unregulated markets. Processing food items at the household level might reduce health hazards from mycotoxins. In this regard, the effect of 3 robust Indian cooking methods (e.g., frying, roasting and pressure cooking) on AFB1 content was evaluated by NRL-NRCG [21]. The processing techniques lowered the AFB1 content to varying degrees (roasting with sodium chloride and citric acid > pressure cooking with sodium chloride and citric acid > frying) at the household level without altering the organoleptic qualities. As the cooking techniques do not require any difficult steps or specialised equipment, they can be easily adopted. This method has been publicised in a science journal [83] and a local Marathi daily newspaper [84]. Currently, NRL-NRCG is appealing to the FSSAI to spread awareness to the public regarding reducing toxin loads in a household setting.

With the rationale of ensuring increased consumer awareness, the FSSAI has also implemented several initiatives, including Eat Right India [85]. The Eat Right India initiative combines three key themes: “Eat Healthy, Eat Safe and Eat Sustainably”. The project aims to increase the demand and supply of safe and nutritious food in a sustainable manner. Because eating habits are formed at a young age, the FSSAI’s Eat Right School initiative (under Eat Right India) aims to educate schoolchildren and, through them, the general public about food safety, nutrition and hygiene [86], whereas Eat Right Campus (another key initiative of the government) ensures the availability of safe food to students and employees at universities [87]. All these projects aim at ensuring balanced nutrition and food security for the Indian population.

As a part of its extended responsibility during the COVID-19 pandemic that impacted all aspects of life [88], the FSSAI went a step ahead to ensure that Indian citizens could procure safe food while staying at home. On this occasion, the agency recommended having a balanced diet [89], considering that the disease is especially harsh on immunocompromised people. The following immunity-boosting foods are enlisted: energy giving cereals and millets; fats or oils; body-building foods (e.g., pulses, eggs, meat, poultry, fish, milk, and milk products) and protective foods (e.g., seasonal fruits and vegetables). Another concerning part of this pandemic was that during the countrywide lockdown, the food supply chain was heavily disrupted [88]. As a consequence, harvested produce from the farms had to be stored for long periods in government warehouses under neglect, with a high probability of developing fungal infections. In a recent study, Kostoff and co-researchers reported that biotoxins such as AFs, OTA and T-2 toxin can harm the immune system and gastrointestinal tract [90], accelerating the chances of COVID-19 contagion [91]. During the pandemic, various non-governmental organisations offered food donations, and the sources and levels of mycotoxin contamination in the food items were unknown. For example, if the food was brought from storage, particularly in rural districts where the supply chain environment supports fungal growth, the population could be exposed to mycotoxin contamination. 

Thus, through these initiatives, the FSSAI proposed hygienic practises for food handling in the kitchen including the purchasing and post-consumption disposal of food leftovers, which might have an overall impact on societal hygienic standards [92,93,94]. Nevertheless, any regulatory implementation is not easy in the country because there are a large number of stakeholders involved in food businesses, and consumers are often less aware of food safety measures. With these campaigns, it is expected that the next generation of consumers will be better prepared in the post-COVID era to protect themselves against microbial food-borne pathogens.

## 7. Implementation Challenges

The implementation of a structured approach to mycotoxin surveillance is not a straightforward process due to several compelling reasons is described below.

### 7.1. Variation in MLs among the Trade-Partner (Exporting and Importing) Countries

The MLs put forward by national and international agencies depend to a large extent upon the lifestyle of a population and its demography. As diets vary across geographic locations, so does the susceptibility to food toxicity, giving rise to diverse perceptions of tolerable health risks [95], and thus differences in MLs. For instance, the ML for patulin in apple juices in India is 50 μg/kg, while it is 10 μg/kg in the EU [55,96]. When the EU implemented the new regulatory rules in the early 1980s, India’s exports of peanut meal to the EU decreased by more than $30 million annually because of more stringent regulations in the EU countries [95]. Economies and business opportunities may be jeopardised, if food law requirements in developed markets are not harmonised with those in emerging markets.

Worldwide, mycotoxin MLs in agricultural commodities are established by the apex authorities, namely the USFDA, FAO/WHO, the European Food Safety Authority (EFSA) and the FSSAI. As these bodies have their own MLs, the following crucial observations can be made, as evident in Table 3. In the case of nuts, India (10 μg/kg) has lower AFB1 MLs than China (20 μg/kg), prohibiting the dumping of items that do not fulfil quality standards. Consequently, India assures that all nuts imported into the country are of the finest quality. However, these two countries share the same ML in cereals (10 μg/kg), particularly rice, indicating a cohesive or equivalent trade system. Similarly, Korea and India may expand their cereals and nuts trade because their AFB1 and total AF MLs are the same, i.e., 10 μg/kg and 15 μg/kg, respectively. Furthermore, going by the overall MLs of AFs, Japan and the EU have a strong hold on Indian imports. If the materials do not meet Japanese and European specifications, revenues may be lowered due to the loss of profit margin and an increase in the costs of operations, fuel and insurance. Surprisingly, Japan and the USA do not possess MLs for AFB1. In this context, India has set superior mycotoxin legislation because it has established MLs for AFB1 and total AFs in a variety of raw and processed commodities.

### 7.2. Disposal of Rejected Lots and Country-Wise Compliance

The disposal of rejected goods at the border poses a matter of concern because there is very little structured surveillance with respect to what should be done with those rejected goods. The rejected export consignments that do not exceed the domestic ML can be imported back after presentation of a No Objection Certificate (NOC) and evaluation by APEDA [43], but other consignments that do not meet the NOC requirements still need to follow a disposal procedure. On certain occasions, items that have passed quality control checks before export fail to comply with MLs in the destination country. Most complaints were reported on shipments to Vietnam, where AF MLs were observed to be much lower at the port of export (15 μg/kg) than the destination port (22–25 μg/kg) [99]. In 2016, Vietnam and the EU returned several shipments to Indian exporters after AF and moisture levels in containers were discovered exceeding specified limits [100]. Such incidents of non-compliance indicate variation during sampling, unhygienic in-transit storage conditions and inter-laboratory analytical variations. This leads to rapid alerts, consignment failures, product recalls and economic losses for stakeholders.

Despite significant efforts to reduce mycotoxin contamination of food and feed products, Indian peanut consignments to EU nations are frequently subjected to border rejections. RASFF notifications of mycotoxins in Indian food and feed products are being investigated. Between 2011 and 2021, 462 notifications were reported for food products, along with 200 notifications for feed. Of these, 91% (419) were for AF in foods. Most of these notifications concerned herbs and spices (49%) in addition to nuts and their products (42%) [64]. NRL-NRCG, EIC and NABL officials investigated such cases and accordingly advised farmers and exporters on preventive and corrective actions.

### 7.3. Toxin Inhomogeneity and Sampling Non-Uniformity

Scientists across the globe agree that sampling is the most crucial step for effective mycotoxin monitoring because it can potentially induce large variations in test results [101]. The high incidence of sampling errors may be attributed to the heterogeneous distribution of AF across the grain and nut lots. Obtaining samples only from grains that are either significantly contaminated or free of mycotoxin will yield inaccurate results [102]. Consequently, APEDA has labelled this variation as a lacuna, and the FSSAI has issued special guidelines for sampling peanuts to detect AF levels [33]. Because fungi do not grow on every kernel in these lots and not all kernels show infection on the surface, it is important to collect a large number of incremental samples at random from the bulk of grains to accurately assess every batch. 

Regardless, there are still hundreds of food products (e.g., mixed spices, botanical products, sauces, Indian cheeses and locally manufactured processed foods) which have not received enough attention from analytical chemists and regulatory scientists. This calls for a meticulous monitoring of mycotoxins in these lesser-studied or even unexplored matrices, as discussed below.

## 8. Understudied and Unexplored Matrices

India is well known for its diverse cultural and culinary habits, and many of its famous cuisines, such as *biryani* (a spiced rice prepared with chicken, mutton or eggs) or *sambar* (a lentil stew with pigeon peas, tamarind broth and spices), use ethnic spice mixes that might vary by region. Earlier, most spices were freshly mixed in homes; however, as industrialisation progressed, people in cities began to rely more on marketed brands of spice mixes. From an analytical perspective, these mixes represent complicated matrices due to the diversity in their profiles of biochemical constituents. Many in-house studies (from NRL-NRCG) have revealed poor recoveries of AFs in these matrices, necessitating optimised sample cleanup and instrumentation methods. In Asian countries, there has been a surge in interest in testing spice mix products for mycotoxin content over the last decade. For example, a study from Malaysia detected AFs in mixed spices used for garnishing fish and meat curries [103]. Another study from Palestine reported that spices (used to flavour chicken curry) were contaminated with AFs ranging from 2.20–8.05 μg/kg [104]. Nevertheless, information on Indian-marketed spice mixes, namely *garam masala* (comprising cardamom, cinnamon, cloves and black pepper), *pav bhaji masala* (made of roasted red chillies, coriander, cloves, cumin, fennel, cinnamon and cardamom) and *sambar masala* (consisting of coriander, cumin, mustard, black pepper, red chillies and fenugreek seeds) are yet unavailable. The Indian regulators must engage in knowledge sharing with these nations and initiate talks of technology transfer to develop indigenous detection methods, which will benefit all of us in the long run.

As dairy products such as cheese, yoghurt and buttermilk are teeming with moisture and nutrients, they are ideal for AFM1 contamination. AFM1 is resistant to the heat and chemical changes that occur during food processing because of their inherent stability, as mentioned earlier [76,77]. According to several studies on bovine milk, conventional processing has no effect on AFM1, as evidenced by the fact that its level in cheese is higher than that in raw milk. To support this, German researchers, Kiermeier and Buchner demonstrated that AFM1 binds to milk proteins and is concentrated in the curd during cheese production [105]. As a result, AFM1 concentrations in cheese are 3–4 fold greater than in milk [105]. Similarly, Indian cottage cheese, commonly known as paneer, has a higher chance of fungal contamination. Until now, there have been only a few studies conducted on AFM1 determination in the Indian context. Using HPLC-FLD, 40 paneer (cottage cheese) samples from Chennai were analysed, in which 43% of the samples demonstrated AF contamination ranging from 0.03–389 μg/kg [106]. In a recent study, 408 milk booth, retailed milk and dairy shop samples were randomly collected in Ludhiana (Punjab) for screening of AFM1 by ELISA. The presence of AFM1 in milk (24.8%), butter (4.1%), cheese (19.2%), and yoghurt (5.8%) exceeded the FSSAI’s limit for the toxin (0.5 μg/kg) [107]. More methods should be established for high throughput testing of Indian paneer, buttermilk and yoghurt from different regions of the country with high accuracy and reproducibility. Public health authorities must routinely test dairy products to minimise the exposure of AFMI to consumers.

In the recent past, Chinese and Korean sauces (e.g., soy sauce) have become popular in India; these sauces were earlier reported to have been contaminated with AFs [108]. Due to the diversity of commercial sauces available in Indian markets, optimising a single extraction method for all of them is difficult. Moreover, sauces and purees are complex matrices composed of fruits, vegetable concentrate, sugar, salt, water, preservatives and colours. Such sauces are common among street meals, and schoolchildren are especially attracted to such roadside savoury dishes. These pigmented, high-moisture food products pose analytical challenges to food chemists. For this, various extraction techniques including solvent partitioning followed by solid phase extraction (SPE), dispersive liquid–liquid microextraction [109], hydrophilic–lipophilic balanced (HLB) and HLB solid phase extraction [110] could be used. Moreover, these matrices must be examined using various instruments, including LC-MS/MS, LC-HRMS and LC-FLD [110]. However, due to financial constraints, not all laboratories have such advanced instruments. This indicates that more public sector funding is required to meet this testing demand.

The study of botanical extracts prepared with variable proportions of raw plant materials such as ginger, spices, kava kava, herbal teas, dried figs, raisins, sultanas and dates needs special attention from the Indian analytical community. Among these, herbal and *masala* (spiced) tea made from the infusion or decoction of herbs, spices or other plant materials (e.g., tea leaves, *tulsi*, *ashvagandha*, ginger, lemongrass and jasmine extract) in hot water are popular beverages among health-conscious Indians. It may be noted that mould contamination of these raw plant materials may lead to a greater mycotoxin load in the final extract. Additionally, relatively more water-soluble mycotoxins (AFs, OTA, ZENs, FUMs) may transfer from the tea matrix into the hot aqueous infusion during brewing [111]. Thus, herbal tea is likely to suffer from matrix interference during analysis. This necessitates the use of a specific method of mycotoxin extraction that reduces matrix influence and maximises recovery. Although a group of researchers in 2020 assessed the dietary exposure to multiple classes of mycotoxin in Indian black and herbal teas [112,113], India, being the world’s second largest tea producer [111], must focus on method development for herbal tea extracts. 

Herbs, like spices, have been used in food preparation for ages to create a distinct flavour in many countries. Dried herbs can be a major source of microbiological risks, especially because they are frequently added to recipes with little processing before consumption [114], for instance, *kasoori methi* (dried fenugreek leaves) and curry leaf powder, among others. They are regularly used in homes, restaurant cuisines and street foods, and their consumption often results in numerous cases of stomach illnesses (reported unofficially), but we still do not have a validated method for testing mycotoxins in them.

India is divided into numerous agro-climatic regions, characterised by unique indigenous foods that play a pivotal role in the cultural ethnicity of its natives. Most of the time, such locally manufactured food items vary in composition. This makes testing and analysis all the more complex because of the non-uniformity of matrices. As of now, most of the testing protocols that are provided in the Indian guidelines have been developed for raw agricultural commodities; however, protocols for street foods have been neglected. Diverse levels and kinds of AFs (B, G and M) have been found in a variety of street foods [115]. The concerning aspect of this is that a large segment of the middle-class Indian population consumes these food products including *bhel* (savoury from puffed rice served with sauces, dried leaves and oils) and *daal badis* (pulse fritters), due to their convenience, taste and affordability. Hence, the FSSAI’s “Eat Right India” movement is spearheading the Clean Street Food Hub Initiative to educate vendors on hygienic practices [85]. Alongside, these projects seek to increase both the demand for and supply of safe and healthful food in a sustainable way through training and capacity building, benchmarking and certification schemes, addressing adulteration and encouraging healthy eating choices.

## 9. Conclusions

This review has outlined how the nodal Indian governmental agencies are assisting the agro-industry stakeholders to progress towards a mycotoxin-free food supply chain. It has also highlighted crucial hurdles they are facing to achieve that goal along with areas for improvement in the existing monitoring system. Despite the geographical conditions of India being conducive to fungal growth, the country has come a long way in setting up a robust export inspection regime with a strong complaint handling system, which has been appropriately covered in this review. Over the years, the government has introduced traceability networks (Peanut.Net) and empowered the regulatory organisations (FSSAI, SB and EIC), which are constantly supervising mycotoxin intervention programmes. Indian testing methods are ever-evolving as per the requirements of the national and international markets, and much progress has been made in this regard. More coordination of governmental activities is required. However, food matrices such as mixed spice products, dry herbs and other street foods should be brought into the mainstream of study, and more scientific attention should be devoted to them. There are enormous opportunities to modernise and automate mycotoxin testing procedures similar to the ones already developed in the case of spices. 

The next few years will be crucial as scientists observe how the changing climate and weather patterns give rise to crop infestations by novel fungal genotypes with elevated resistance to disinfection. As far as administration is concerned, the governmental agencies and industries need to follow a systematic approach to the disposal mechanism of rejected consignments at the production units or the geographical border. Trading countries need to work out amicable ways to strike a balance in prioritising between health benefits and financial consequences; only then can we achieve economic progress and ameliorate the health status of Indian citizens and the world. We recommend adopting robust agricultural practices to arrest mycotoxin spread on farms rather than investing in detoxification attempts. Following the COVID-19 pandemic, Indian agencies implemented a variety of recommendations for a balanced diet to strengthen immunity. The downward trend of recorded AF-related food contamination notifications for the products of India origin indicates the efforts made by APEDA, FSSAI, EIC, NRCG, SB and other stakeholders to educate and raise awareness of primary production controls, GAPs among farmers, improve quality control and testing and eventually write the success story of mycotoxin monitoring, regulation and analysis.

## Figures and Tables

**Figure 1 foods-12-00705-f001:**
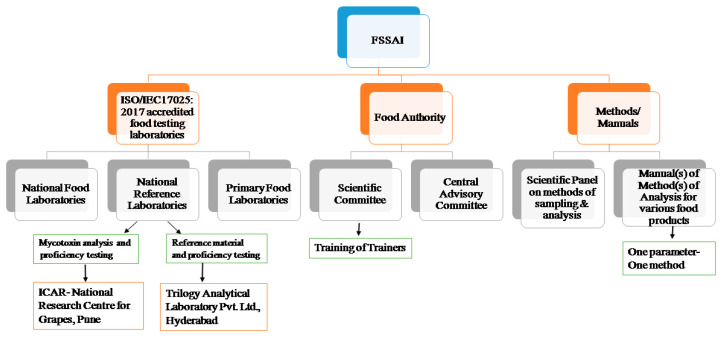
Schematic view of role of FSSAI in capacity building trough food testing laboratories, scientific community, and analytical methods to combat mycotoxin spread in India.

**Table 1 foods-12-00705-t001:** Current FSSAI regulatory limits for mycotoxins in India.

Mycotoxin	Food Product	FSSA(I) Regulatory Limit (μg /Kg)
**AFs**	Cereal and Cereal Products	15
	Pulses	15
	Nuts	
	Nuts for further processing	15
	Ready to eat	10
	Dried figs	10
	Oilseeds or oil	
	Oilseeds for further processing	15
	Ready to eat	10
	Spices	30
	Betelnut/Arecanut	15
**AF M1**	Milk	0.5
**OTA**	Wheat, barley and rye	20
**PAT**	Apple juice and Apple juice ingredients in other beverages	50
**DON**	Wheat	1000

**Table 2 foods-12-00705-t002:** Summary of currently practised mycotoxin detection methodologies as per FSSAI Manual of Methods of Analysis of Foods—Mycotoxins.

Mycotoxin	Matrix	Detection Method	LOD	LOQ	Linearity Range
**Total AFs**	Groundnuts and groundnut products, oilseeds and food grains	Thin-Layer Chromatography (TLC)-Ultraviolet (UV)	-	-	-
**Total AFs**	Food and feed	TLC-UV: Romer Mini Column Method	-	Almonds ≥ 5 ng/g, corn, cotton seed meals, nuts ≥ 10 ng/g, mixed feeds ≥ 5 ng/g	-
**Total AFs**	Corn and groundnuts	TLC-UV	-	B1, G1 = 5 to 50 ng/g, B2, G2 = 3 to 15 ng/g	-
**Total AFs**	Corn and peanut powder/butter	HPLC-FLD (Derivatised)	0.3 ng/g	-	B1, G1 = 0.25 to 2 ng/mL, B2, G2 = 0.125 to 1 ng/mL
**AFs (B1, B2 and G1)**	Corn, cottonseed, peanut, peanut butter	ELISA (Immuno-dot screen cup)	-	Cottonseed, peanut ≥ 20 ng/g, corn, raw peanuts ≥ 30 ng/g	-
**AFs (B1, B2 and G1)**	Corn	ELISA (Afla-20 cup test)	-	≥20 ng/g	-
**AF B1 and Total AFs**	Peanut butter, pistachio paste, chilli, paprika powder and other spice powders, dried figs and other dried fruits	HPLC-FLD (IAC cleanup and derivatised)	-	-	B1, G1 = 0.4 to 3.6 ng/mL, B2, G2= 0.08 to 0.72 ng/mL
**AF B1**	Baby food	HPLC-FLD (IAC cleanup and derivatised)	-	≥0.1 ng/g	0.01 to 0.07 ng/mL
**Total AFs +** **their derivatives (G2a, B2a)**	Olive oil, peanut oil and sesame oil	HPLC-FLD (IAC cleanup and derivatised)	-	2 to 20 μg/kg	B1 = 0.4 to 10 ng/mL, G1 = 0.2 to 5 ng/mL, B2, G2 = 0.1 to 2.5 ng/mL
**Total AFs**	Peanut, peanut products and cereals	UHPLC-FLD (Direct method)	-	0.008 μg/kg for B1 and G1, 0.003 μg/kg for B2 and G2	0.02 to 10 ng/g
**AFs M1 and M2**	Milk	LC-FLD	-	-	-
**AF M1**	Milk	LC-UV Vis	-	˃0.02 ng/mL	0.05 to 1 ng/mL
**Total AFs**	Matrices other than described above	HPTLC-UV	-	-	-
**DON**	Wheat	TLC-UV	-	≥300 ng/g	-
**OTA**	Barley	TLC-UV	-	-	-
**OTA**	Barley	HPLC-FLD(IAC Cleanup)	0.1 ng/g	˃1 ng/g	0.5 to 10 ng/mL
**AF + OTA**	Cereals	UHPLC-FLD	AF = 0.02 ng/g, OTA = 0.1 ng/g	-	0.02 to 10 ng/mL for AF, 0.1 to 10 ng/mL

**Table 3 foods-12-00705-t003:** MLs of AFs (B1 and Total) in various food commodities traded from India and major trading countries.

Country/ Region	Foodstuff	AFB1 (μg/kg)	Total AF (μg/kg)	Reference
India	Cereals and Nuts Spices	10 15	15 30	[96]
European Union	Peanut and their products Rice Spices	2 5 5	4 10 10	[55]
Japan	All foods	10	-	[97]
USA	All foods except milk	-	20	[95]
China	Peanut and their products Rice	20 10	- -	[98]

## Data Availability

The data presented in this study are available on request from the corresponding author.

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
