# Peer review of "Mycotoxin Monitoring, Regulation and Analysis in India: A Success Story"

_foods, 2023, doi:10.3390/foods12040705_

Round 1

Reviewer 1 Report (Previous Reviewer 1)

The authors put a lot of effort into completing and rewriting this paper. It offers considerable information on current Indian framework for the management of Mycotoxins. However, I cannot resist making a few small comments.

Line 196: unnecessary space

Line 287: The authors wrote there are 12 NFLs, but this NFLs are located in 10 different places. So I am assuming that there are 2 places that have 2 NFLs (or 1 with 3). Am I correct? Unless the two ancillary laboratories located in Chennai and Kolkata are included in the 12 original NFLs?In both cases, an explanation would be necessary.

Figure 1: Central Adversary Committee (with capital letters?)

Lines 376-379: “Following this, in 2021, there was only 1 notification for OTA in rice, 7 notifications for AFs in herbs and spices and 14 notifications for AFs in peanuts and their products. This indicates that there is a steady improvement in mycotoxin management in fields (pre-harvest) and pack houses (post-harvest).”

Considering that 2021 was during the covid pandemic, I would not be so sur that this decrease in number was due to an improvement in mycotoxin management and I would simply remove these sentences.

Line 399-401: I would like to know if these laboratories are included in the FSSAI laboratories or if they are completely different.

Lines 686-699: The first paragraph related to AFM1 detection in milk is out of scope with the broader title of the section: 5. Evolution of mycotoxin analysis techniques. It should be removed (at least Lines 691-699) and the title for 5.6 Methods developed to combat AFM1 toxicity should be modified for 5.6 AFM1 detection in milk.

Line 779: The MLs may vary from a country to another not because the population is more or less resistant to food poisoning but mostly because the diet is not the same in all countries!

Lines 783-785 should be removed. I do not understand why authors decide to point the finger at developed countries, but it is not helpful.

Author Response

Reviewer 1:

  1. Line 196: unnecessary space

Response: Removed unnecessary space.

  1. Line 287: The authors wrote there are 12 NFLs, but this NFLs are located in 10 different places. So I am assuming that there are 2 places that have 2 NFLs (or 1 with 3). Am I correct? Unless the two ancillary laboratories located in Chennai and Kolkata are included in the 12 original NFLs? In both cases, an explanation would be necessary.

Response: There are 11 NFLs located at 9 different places and one location i.e. Kochi have 3 laboratories. Additional two laboratories as ancillary laboratories are located in Chennai and Kolkata. Please see Page 6 and 7.

  1. Figure 1: Central Adversary Committee (with capital letters?)

Response: Corrected as suggested. It is Central Advisory Committee.

  1. Lines 376-379: “Following this, in 2021, there was only 1 notification for OTA in rice, 7 notifications for AFs in herbs and spices and 14 notifications for AFs in peanuts and their products. This indicates that there is a steady improvement in mycotoxin management in fields (pre-harvest) and pack houses (post-harvest).”

Considering that 2021 was during the covid pandemic, I would not be so sure that this decrease in number was due to an improvement in mycotoxin management and I would simply remove these sentences.

Response: Noted and removed as suggested

  1. Line 399-401: I would like to know if these laboratories are included in the FSSAI laboratories or if they are completely different.

Response: These are ISO-17025 accredited, APEDA approved laboratories, which have access to Peanut.Net. These laboratories are also recognised by FSSAI, but their role in Peanut.Net is typically focused on export.

  1. Lines 686-699: The first paragraph related to AFM1 detection in milk is out of scope with the broader title of the section:  Evolution of mycotoxin analysis techniques. It should be removed (at least Lines 691-699) and the title for5.6 Methods developed to combat AFM1 toxicity should be modified for 5.6 AFM1 detection in milk.

Response: Modified the title as 5.6 AFM1 detection in milk. The first paragraph mentioned about the seriousness of AFM1 toxicity and necessity of method development to cope up with lower MLs set by different regulatory bodies.

  1. Line 779: The MLs may vary from a country to another not because the population is more or less resistant to food poisoning but mostly because the diet is not the same in all countries!

Response: Noted and modified as suggested (Section 7.1).

  1. Lines 783-785 should be removed. I do not understand why authors decide to point the finger at developed countries, but it is not helpful. Response: This part has been modified. Please see the highlighted portion of 7.1 in page 19.

Reviewer 2 Report (Previous Reviewer 2)

Foods

Manuscript ID: foods- 2141757

Type of manuscript: Review

Section: Food toxicology

Title: Mycotoxin Monitoring, Regulation and Analysis in India: A Success Story

Comments to the Author 

Overall, this is an interesting study, well written and structured. In this study, the authors aim to promote a systematic picture of the role played by two major governmental agencies in the country for mycotoxin control at the domestic level and international trade. I think that the data are informative and the experimental designs are no major flaws.

Author Response

Reviewer 2:

Comments to the Author 

Overall, this is an interesting study, well written and structured. In this study, the authors aim to promote a systematic picture of the role played by two major governmental agencies in the country for mycotoxin control at the domestic level and international trade. I think that the data are informative and the experimental designs are no major flaws.

Response: We appreciate this comment and recommendation.

Reviewer 3 Report (New Reviewer)

This manuscript describes the progress of research, government regulatory measures and legislation in India in the areas of monitoring, risk assessment and hazard control of mycotoxin contamination in domestic and exported food products. It also summarises the challenges and future directions in the field of mycotoxins. It is of great importance for safeguarding food safety and human health. The manuscript fits within the scope of the journal.

I have some major recommendations for authors:

1 It is recommended to reduce the number of keywords, preferably to no more than 5.

2 This paper provides an overview of the contamination of AFs and OTA in major export foods such as peanuts, nuts and spices. Whether other types of mycotoxins in other cereals (e.g. wheat, maize) have been analysed for contamination, in particular the contamination of domestic food products with multiple mycotoxins, is important for risk assessment and the development of management measures.

3 Manual of detection methods mentioned in this paper, which mycotoxins are specifically included, only DON、ZEN、AFs、FUM、OTA and PAT? Does it include derivatives and modifications such as 3ADON, D3G, ZAL, Z14G, FB2/3, HFB1, T-2/HT-2. It is recommended that the name of the included mycotoxin, the detection method and the specific parameters (e.g. detection limit, linear range) are listed in a table.

4 Has the Indian government established limits for certain mycotoxins in some foods? Only the limits for AFs are shown in this article and it is recommended that other limit values are also shown in the appropriate places.

5 Measures for mycotoxin contamination control are described in this paper, but there are few specific measures described in the paper (e.g. through different cooking methods) and it is suggested that a table listing the control methods currently available or governmental control norms and the effectiveness of mycotoxin reduction etc. be included. As biological control methods (microbial antagonism or degradation) have become a hot topic of research in recent years and are considered to be a good quality method for mycotoxin contamination control, could the progress of research on biological control methods by the Indian government and related discussions be added.

Author Response

Reviewer 3:

It is recommended to reduce the number of keywords, preferably to no more than 5.

Response: Modified as suggested

  1. This paper provides an overview of the contamination of AFs and OTA in major export foods such as peanuts, nuts and spices. Whether other types of mycotoxins in other cereals (e.g. wheat, maize) have been analysed for contamination, in particular the contamination of domestic food products with multiple mycotoxins, is important for risk assessment and the development of management measures.

Response: Methods of analysis in cereals namely rice, millets and corn (maize) have been briefed in sections 5.1 and 5.4. The mycotoxins other than AFs and OTA are also monitored, but we did not come across published literature to cite. Please see Table 1, in which the ML of DON in wheat is mentioned.

  1. Manual of detection methods mentioned in this paper, which mycotoxins are specifically included, only DON、ZEN、AFs、FUM、OTA and PAT? Does it include derivatives and modifications such as 3ADON, D3G, ZAL, Z14G, FB2/3, HFB1, T-2/HT-2. It is recommended that the name of the included mycotoxin, the detection method and the specific parameters (e.g. detection limit, linear range) are listed in a table.

Response: Only derivatised forms of AFB1 (B2a) and AFG1 (G2a) are mentioned in the manual. Added a separate table summarizing mycotoxins, matrices, detection methods, and linearity.

  1. Has the Indian government established limits for certain mycotoxins in some foods? Only the limits for AFs are shown in this article and it is recommended that other limit values are also shown in the appropriate places.

Response: MRLs of other notable mycotoxins have been included in a separate table (Table 1) as suggested.

  1. Measures for mycotoxin contamination control are described in this paper, but there are few specific measures described in the paper (e.g. through different cooking methods) and it is suggested that a table listing the control methods currently available or governmental control norms and the effectiveness of mycotoxin reduction etc. be included. As biological control methods (microbial antagonism or degradation) have become a hot topic of research in recent years and are considered to be a good quality method for mycotoxin contamination control, could the progress of research on biological control methods by the Indian government and related discussions be added.

Response: The mentioned content has been included in section 3. Here we have elaborated some government on farm-control strategies like GAP, GMP, GSP and GHP, along with pre-and post-harvest guidelines by FSSAI. Biological control of mycotoxin producing fungi has also been touched upon, in section 3 and is highlighted.

Round 2

Reviewer 3 Report (New Reviewer)

Dear author

This and the manuscript have been revised according to the comments.

This manuscript is a resubmission of an earlier submission. The following is a list of the peer review reports and author responses from that submission.

Round 1

Reviewer 1 Report

This is an interesting manuscript about the Indian framework for the control of mycotoxins in food commodities. The paper is well written but badly structured and I strongly suggest the reshape of several sections.

The objectives of the review are unclear. It seems that the authors want to present the regulatory framework of India but in the end some “parasites” sections make it very unclear and the section on methods is too long and really looks like a list of methods despite the fact that authors state the opposite in the introduction.

1.       Introduction:

L34 There is an extra space before the word “So”

L32-36: I would prefer to link genera and mycotoxins, rather than making two separate lists

L63 There is an extra space before the word “Small”

L66 There is an extra space before the word “In North Indian”

L69 There is an extra space before the word “The production”

L75 extra space between 10 and hazards

L77 Extra space before “Way back”

L90-96 on spices. This paragraph is not necessary at this point of the introduction, and it feels like it was dropped here randomly.

-          Why do you want a full paragraph focused on spices?

The objectives of the review are clear, but I would remove the following sentence (This review is not intended to be an exhaustive list of mycotoxin analytical methods publications) or rewrite it to mix it with the next one.  

2. Methodology

L138-155: After a first reading of the paper, it seems that several “relevant data” are not included in the review, especially for the following two:

a. Understand how monitoring and controlling mycotoxins in food influences public health.

b. Determine why Indian agricultural practices are prone to fungal infection and generation of mycotoxins in grains, nuts, spices and milk  

L152 Extra space before “provide”

L163-176. This list seems unnecessary

L 178 Inclusion Criteria: I would like to know the period of time that was covered

L183 Exclusion criteria: What about Quality Assurance ? 

3. India’s food safety framework for domestic surveillance of mycotoxin

Some questions remain after this section:

-          Do India have surveillance programmes for domestic surveillance?

-          Do India have a risk assessment process to prioritize food commodities that have to be monitored?

-          What about other aspects of domestic surveillance (risk communication, risk management, etc…?)

L224-230 APEDA seems to be for exports only so this part seems to be out of the scope of this section

L242 Table 1 is confusing. I understand that it is the MLs that are used to trade from India to other countries as stated in the title of the table. But then why India is the major trading country for the trade of cereal and nuts, and spice?

And this paragraph is supposed to be about domestic surveillance framework, so I do not understand why the article present MLs for international trade. If it would be for import, I would understand but it is not written, and it is not clear if it is for this context or not.

L253-267 A figure would be appreciated to have a global mapping of the regulatory framework

L254 and 255. Why did you use {} ? (not important)

L272 I would remove the sentence During the fiscal year 2019-20, 21 subject specific scientific panels were operational, which held a total of 67 meetings [40].

This is purely numbers to write numbers because the reader cannot know what the scientific panels and the subjects of the 67 meetings are. What is the information that the authors want to bring here? 

4. Mycotoxin surveillance for export facilitation

L295-313 It is not explained why the authors focus on Europe specifically, but I am confident that several other countries also have a high level of safety.

However, the part with the RASFF numbers is clear and I do understand why the authors decided to focus on spice and peanut after that, but it is not clear if India only have programmes for these food commodities or if others are concerned but not included in this review.

Furthermore, the end of the paragraph is confusing (“In response to an increasing number of RASFF notifications of AF contamination in peanuts imported from India, the EC/Directorate-General for Health and Food Safety has recently conducted certain inspections, which indicate a steady improvement in implementing the mycotoxin management practices at field (pre-harvest) and pack-houses (post-harvest).”) because I am not sure what are the numbers and the period of time is not mentionned. In addition, the number that are given just before this sentence are in contradiction with this statement (it looks like there is a decrease of RASFF alerts instead of an increase)

L314 “established” is used twice in the same line

L317-322 I would like to know if these food commodities have been chosen after a risk assessment or if it was motivated by financial impacts only.

L330 Why 14 laboratories just for Indonesia??

L362 extra space between testing and plant

L379 What about noncompliant laboratories?

L384 to 424: this whole sub-section seems to be out of scope considering the main title of this section.

-          First paragraph is about commercial aspect of spice

-          Second paragraph is about quality improvement but there is no link done with impact on mycotoxins or exportation directly

-          Third paragraph is in the scope but very vague and it does not provide specific information (vague)

L389 extra space

L391-394: The EU has established the MLs for AFs in imported spices as already stated. Many times, Indian authorities have been perplexed as to why, despite meeting all official export inspection requirements, consignments are rejected in the EU.

I do not feel comfortable with this sentence. I would suggest removing the subjective point of view and stay neutral. 

5. Evolution of mycotoxin analysis techniques

In general, this section is useful and perfectly clear. I do not understand why the authors decided to separate section 5 and 6.

L436 Why the Indian laboratories limited the application of the EU regulation to peanut and peanut related products (why not others?)

L440 random bolt point.

L454-463 How many methods have been accepted? 

6. Make in India analytical methods

This whole section is way too long, could be resumed in one single table and included in the precedent section. I would also remove subsection 6.4 “Mycotoxin detoxification…” that seems not relevant in a regulatory context.  

L467 Rephrase: Now, we discuss..

In this paragraph, we will discuss… 

L493 -497

Why exactly are you talking about results from Doha (Qatar)? What is the link with India? I would suggest removing this part: “Contamination of spices with AFs could be a potential hazard for health of consumers in India [54]. AFs were found in five spices {turmeric, black pepper, garam masala (a blend of ground spices), chilli and tandoori masala} in a research on marketed spice samples in Doha, and except for garam masala, the other four varieties surpassed the MLs for AFB1 and/or total AF. The presence of fungal propagules was the highest in chilli powder [55]. For this, NRL-NRCG has been doing its best to develop analytical methods to detect mycotoxins in spices.” 

7. Concerns in mycotoxin control

This section is unnecessary, and I would remove it entirely (no regulatory context). The scientific content seems perfectly fine but is simply out of scope. 

8. Implementation challenges

L641-643: “Such disparities in food laws between advanced and emergent markets lead to loss in economy and trade opportunities for the latter.”

It is maybe true, but this sentence should be removed or rephrased. Nowadays, MLs are setup after a robust risk assessment process and in accordance with the Codex guidelines, and should not be based on “trade opportunities”. To be honest, I am pretty sure that it was not the case in the 80s but the sentence does not provide any nuance.

L645 “In the absence of vigilance, these consignments are often released into the domestic markets at a competitive price, thereby creating public health alarm”.

I would request a reference for this kind of statement.

L647 remove “often”

L653-657: very similar to 305-313 (same info)

L670-674: “Nevertheless, global regulatory bodies such as Codex and the EU also do not have provisions for sampling and analysis for such locally manufactured foods. Even if methods were to be developed, this would require huge data generation and survey. The dearth of pertinent toxicological data is also a significant hindrance to the appropriate testing of mycotoxins in food and feed.”

I agree with the whole paragraph but still find it awkward from a regulatory standpoint. Why exactly should India wait until Codex or EU provide guidance to do something? It looks to me that if India identifies an issue in a framework, it is India that should provide a solution. In addition, the Indian context is very different to the EU context, and it seems very unlikely to me that EU will provide a provision for that kind of sampling considering that EU will never be in contact with “locally consumed products” from India.

Then, regarding the amount of data needed and survey, I totally agree with the statement, but it is written in a way that make it looks like it is a reason for doing nothing which seems weird considering the context. Consequently, I would rephrase this whole section. 

9. Conclusion

L691 Conclusion is paragraph number 9 instead of 8

Reviewer 2 Report

Manuscript ID: foods-2019534

Type: Review

Title: Mycotoxin Monitoring, Regulation and Analysis in India: A Success Story

Comments to the Author 

Overall, this is an interesting study, well written and structured. In this study, the authors aim to promote a systematic picture of the role played by two major governmental agencies in the country for mycotoxin control at the domestic level and international trade. I think that the data are informative.

Reviewer 3 Report

An interesting review dealing with monitorization, regulation and analysis of mycotoxins in India. I consider that this manuscript could be minor revision.

In order to make the text easier to read I would recommend the inclusion of a list of abbreviations. In this regard, I have been able to detect the use of different abbreviations for the same concept. For example, FLR and FLD. Please check throughout the manuscript for this and all other abbreviations. In addition, the full term is sometimes used when the abbreviation has already been defined (for example in line 351 for FLD). By the way, has MS/MS abbreviation been defined previously to line 351?

Line 39: “AFB1 has been identified as a group I carcinogen”. Perhaps better: AFB1 has been catalogued within the group I carcinogen by…

Methodology. I recommend a restructuring of this section. Although much of the information provided is valuable, I believe that a journal of the prestige of Foods should reconsider the need of its inclusion in the way it is currently approached.

Lines 224-230. Please, consider dividing this sentence into two or more, in order to make it easier to read.

Table 1. Please consider widening the width of the second columna (Foodstuff).

I would recommend to omit sentence in lines 422-424.

Section 6. Please, rewrite the title of this section.

Line 576-577: The EU guidelines direct that barring AFM1, none of the other AFs are found in milk. Please rewrite.

Section 6.4. The true significance of cooking techniques in reducing the level of mycotoxin contamination needs to be indicated in this section, as the reader is not aware of the reality until paragraph 7.

Line 650. What do you mean by analytical differences?

References: Please, adapt the bibliography to the format of the journal.

Reference 52. Please, check year. I think is 2018 instead of 2017.